# Evaluation of Quantitative Ga-68 PSMA PET/CT Repeatability of Recurrent Prostate Cancer Lesions Using Both OSEM and Bayesian Penalized Likelihood Reconstruction Algorithms

**DOI:** 10.3390/diagnostics11061100

**Published:** 2021-06-16

**Authors:** Mark J. Roef, Sjoerd Rijnsdorp, Christel Brouwer, Dirk N. Wyndaele, Albert J. Arends

**Affiliations:** 1Department of Nuclear Medicine, Catharina Hospital Eindhoven, Michelangelolaan 2, 5623 EJ Eindhoven, The Netherlands; christel.brouwer@catharinaziekenhuis.nl (C.B.); dirk.wyndaele@catharinaziekenhuis.nl (D.N.W.); 2Department of Medical Physics, Catharina Hospital Eindhoven, Michelangelolaan 2, 5623 EJ Eindhoven, The Netherlands; srijnsdorp@outlook.com (S.R.); bertjan.arends@catharinaziekenhuis.nl (A.J.A.)

**Keywords:** repeatability, Ga-68 PSMA PET/CT, Bayesian penalized likelihood reconstruction, prostate cancer

## Abstract

Rationale: To formally determine the repeatability of Ga-68 PSMA lesion uptake in both relapsing and metastatic tumor. In addition, it was hypothesized that the BPL algorithm Q. Clear has the ability to lower SUV signal variability in the small lesions typically encountered in Ga-68 PSMA PET imaging of prostate cancer. Methods: Patients with biochemical recurrence of prostate cancer were prospectively enrolled in this single center pilot test-retest study and underwent two Ga-68 PSMA PET/CT scans within 7.9 days on average. Lesions were classified as suspected local recurrence, lymph node metastases or bone metastases. Two datasets were generated: one standard PSF + OSEM and one with PSF + BPL reconstruction algorithm. For tumor lesions, SUVmax was determined. Repeatability was formally assessed using Bland–Altman analysis for both BPL and standard reconstruction. Results: A total number of 65 PSMA-positive tumor lesions were found in 23 patients (range 1 to 12 lesions a patient). Overall repeatability in the 65 lesions was −1.5% ± 22.7% (SD) on standard reconstructions and −2.1% ± 29.1% (SD) on BPL reconstructions. Ga-68 PSMA SUVmax had upper and lower limits of agreement of +42.9% and −45.9% for standard reconstructions and +55.0% and −59.1% for BPL reconstructions, respectively (NS). Tumor SUVmax repeatability was dependent on lesion area, with smaller lesions exhibiting poorer repeatability on both standard and BPL reconstructions (F-test, *p* < 0.0001). Conclusion: A minimum response of 50% seems appropriate in this clinical situation. This is more than the recommended 30% for other radiotracers and clinical situations (PERCIST response criteria). BPL does not seem to lower signal variability in these cases.

## 1. Introduction

In western Europe, United States and Canada, prostate cancer has the highest incidence of all cancers in men, with a global incidence estimated at over 1.6 million in 2015 [1]. Nearly half of the patients will experience a relapse during their lifespan, either locally in the prostate (bed) and/or in lymph node or bone metastases [2]. Biochemical relapse (BCR) is defined as a PSA relapse after initial treatment, with different definitions for BCR after radical prostatectomy or radiotherapy (i.e., external beam radiotherapy or brachytherapy) [3]. Management strategies in BCR are focused on early detection of disease, with patients with low volume disease having the best prognosis [4]. Patients with low volume disease may have only a few lesions of small size that make them qualify for focal therapy, e.g., stereotactic body radiotherapy (SBRT), with the intent of postponing systemic therapy [5]. Focal therapy can thus be directed against either local relapses or against metastases (metastasis directed therapy, MDT). PSMA PET/CT is a widely accepted imaging modality in BCR, showing lesions with high contrast to their background [6]. Using PSMA PET/CT for monitoring of focal therapy to these lesions is generally less well accepted but promising [7].

A standardized quantitative approach still needs to be developed, however.

It is known that uptake measurements of radiolabeled tracers with PET in vivo suffer from many inaccuracies, as demonstrated by experience with FDG [8], and that this requires evaluation and standardization prior to application as response parameter [9]. This probably applies equally to PSMA-PET expression. Before quantification of PSMA, expression can be used as a surrogate biological parameter to identify response to treatment, and before we can design sufficiently powered response evaluation studies, we need to know the characteristics of the measurement technique. An important factor that is currently less known is the normal day-to-day variability in Ga-68 PSMA expression of tumor, i.e., repeatability.

The main aim of this pilot study is to formally determine the repeatability of Ga-68 PSMA in both relapsing and metastatic tumor, with focus on small lesions. In small lesions, partial volume effects become increasingly relevant. Partial volume effects start to play a role when lesion size falls below 2 to 3 times the spatial resolution of the PET system defined by its full width at half maximum (FWHM), and are expected to increase signal variability [10,11].

The spatial resolution that can be achieved in PET imaging is limited by physical characteristics such as positron range and noncollinearity of annihilation photons depends furthermore on scanner properties such as detector size and geometry, acquisition parameters and on the reconstruction method applied.

For 18F-fluordeoxyglucose (FDG), Rogasch et al. have demonstrated that the Bayesian penalized likelihood (BPL) reconstruction algorithm called Q.Clear (GE Healthcare, Milwaukee, WI, USA) can yield consistently improved reconstructed spatial resolution at high and medium signal to background ratios, compared to OSEM +PSF + TOF and OSEM + TOF [12].

Q.Clear is an iterative reconstruction algorithm that runs to full convergence by controlling the noise introducing a relative difference noise penalty [13,14]. The noise control term is a function of the signal values in neighboring voxels and is controlled by a unitless penalization factor (the beta value), which is the only user-input variable set in the algorithm [14]. In solitary pulmonary nodules (SPNs), BPL is claimed to improve lesion visibility, when compared to OSEM [15,16]. We hypothesized that in small prostate cancer lesions, which also may comprise of only 2 or 3 voxels, the noise penalty of the BPL algorithm would improve uptake quantification to the voxel level, thus lowering signal variability. This is the second aim of our study.

## 2. Materials and Methods

### 2.1. Patients

Thirty patients with biochemical recurrence of prostate cancer scheduled for routine clinical Ga-68PSMA PET/CT were prospectively enrolled in this pilot study (NL52809.100.16/R16.058/Ga-68 PSMA test-retest study) between January 2018 and July 2019. Their scans were screened for evaluable PSMA-positive tumor lesions by two board certified nuclear physicians (MJR and CB). Seven patients had no evaluable tumor lesions and were excluded from the study. The remaining twenty-three had their second (retest) scan within on average 7.9 days (range 6 to 23) of their initial (test) scan and were evaluated for measurements in assigned tumor lesions. Only one patient had a relatively long interval of 23 days between test and retest due to logistical issues, where the next highest value was 10 days. Tumor lesions were classified as suspected local recurrence, lymph node metastases or bone metastases. Both baseline scans were performed before any treatment had begun. The study was approved by the local institutional ethics review board, and all patients had given written approval before any scanning was done (NL52809.100.16/R16.058/Ga-68 PSMA test-retest study, date of approval 22 March 2017).

PET images were acquired on a PET/CT scanner (GE Healthcare Discovery 710), with a 2.5 min acquisition per bed position, on average 58 (range 55 to 69) min after injection of on average 1.4 (range 0.9 to 1.7) MBq/kg dose of Ga-68 PSMA. Two datasets were reconstructed: one standard ordered subset expectation maximization (OSEM) reconstruction including both time-of-flight (TOF) and point spread function (PSF) modelling and a second one using the Bayesian penalized likelihood (BPL) algorithm, called Q.Clear [17]. For the OSEM reconstruction, 2 iterations and 24 subsets were used. BPL reconstruction also included TOF and PSF. BPL (including TOF and PSF) is subsequently referred to as BPL, same as for OSEM. A matrix size of 256 × 256 was used, resulting in voxels of 2.73 × 2.73 × 3.27 mm^3^. With respect to filtering, a 6.4 mm Gaussian filter and 1:4:1 filter in axial direction were applied. Both used low dose CT for attenuation correction. For the BPL algorithm, a beta value of 600 was used. This value was found optimal in a Ga-68 PSMA phantom study using spheres of 5–37 mm diameter (this special issue).

### 2.2. PET and CT Analysis

For both the test and the retest datasets, the PET and low-dose CT images were processed independently. Imaging reading was performed using dedicated software for PET/CT imaging (Philips IntelliSpace Portal 9.0, Eindhoven, The Netherlands).

In PET, tumor lesion size was measured in the axial plane using a fixed PET windowing upper level (UL) of 10 used for stretching of the greyscale. Both long and short axis were measured; lesion area was calculated according to the simple formula for round and oval lesions: A = π × half long axis × half short axis. Within this area, the pixel with the highest standardized uptake value is designated the SUVmax (injected dose/kg body weight). Thus SUVmax was measured in all tumor lesions. The small size of most of the lesions did not allow for measurement of other meaningful SUVs such as SUVpeak that need lesions of at least 1 cm^3^. Low dose CT was used to check for appropriateness of the lesion area measured in PET if possible (i.e., with the exception of some bone metastases not visible on CT).

### 2.3. Statistical Analysis

The repeatability of SUVmax in tumor lesions was assessed with Bland–Altman analysis by reporting the mean (bias) and limits of agreement (defined as mean ± 1.96 SD) of the differences between the two measurements of individual lesions. Bland–Altman analysis was preferred over intra class correlation coefficients on the basis of previous recommendations [18,19]. Assessment of signal variability between lesion areas and SUVmax was assessed with F-testing.

For the comparison of the two reconstruction techniques a paired t-test was used, i.e., BPL versus standard OSEM.

## 3. Results

A total number of 65 PSMA-positive tumor lesions were found in 23 patients (range 1 to 12 lesions), see Table 1 for patient characteristics. In addition, in 7 patients no lesions were found, and therefore, these were excluded. In theory, a perfect test–retest will result in identical values for the test and the retest. In daily practice, however, measurements of a lesions SUVmax in repeated acquisitions will yield results normally distributed around the true value. In tumor lesions showing an increased SUVmax from test to retest, the average increase was 16.8% ± 10.6% (SD) on standard OSEM reconstructions (33 lesions) and 22.1% ± 18.6% (SD) on BPL reconstructions (30 lesions). In tumor lesions showing a decreased SUVmax from test to retest, these values were −21.0% ± 15.2% (SD) on standard OSEM reconstructions (32 lesions) and −23.2% ± 19.7% (SD) on BPL reconstructions (35 lesions), respectively. Overall repeatability in 65 lesions was −1.5% ± 22.7% (SD) on standard OSEM reconstructions and −2.1% ± 29.1% (SD) on BPL reconstructions. The small difference between both reconstructions was not statistically significant.

As shown in Figure 1A, overall repeatability of SUVmax had upper and lower limits of agreement of +42.9% and −45.9% for standard OSEM reconstructions and +55.0% and −59.1% for BPL reconstructions, respectively. For suspected local recurrence, SUVmax had a repeatability of −5.0% ± 14.4%, with upper and lower limits of agreement of +23.2% and −33.1% for standard OSEM reconstructions. For BPL reconstructions, SUVmax repeatability was −9.5% ± 16.8%, with upper and lower limits at +23.5% and −42.5%. See Figure 1B. For suspected lymph node metastases, SUVmax repeatability was −4.5% ± 22.8% for standard OSEM reconstructions, with upper and lower limits of +40.1% and −49.1%. For BPL reconstructions, SUVmax repeatability was −3.3% ± 30.6%, with upper and lower limits at +56.6% and −63.2%. See Figure 1C. For suspected bone metastases, SUVmax had a repeatability of +4.4% ± 26.1%, with upper and lower limits of agreement of +55.7% and −46.8% for standard OSEM reconstructions. For BPL reconstructions, SUVmax repeatability was +2.7% ± 32.8%, with upper and lower limits at +67.0% and −61.6%. See Figure 1D. None of the differences between standard OSEM and BPL reconstructions were statistically significant.

Tumor SUVmax repeatability was dependent on lesion area, with smaller lesions exhibiting poorer repeatability on both standard OSEM and BPL reconstructions (F-test, *p* < 0.0001). See Figure 2A,B.

Tumor absolute SUVmax was higher in BPL reconstructions than in standard OSEM reconstructions for all lesions (data not shown). The relative increase in measured SUVmax in the BPL reconstructions (as compared to standard OSEM reconstruction) was dependent on lesion size. Smaller lesions (lesion area < 200 mm^2^) showed a significant larger increase of SUVmax as compared to larger lesions (lesion area > 200 mm^2^): 44.3% ± 4.6% versus 25.5% ± 42.2% for the test scans (*p* = 0.004) and 43.5% ± 3.9% versus 18.6% ± 3.1% for the retest scans (*p* < 0.001), respectively. See Table 2.

## 4. Discussion

In this prospective test–retest study, we formally report the repeatability of Ga-68 PSMA in both relapsing and metastatic tumor. The main finding of this study is the relatively high day-to-day variability of tumor SUVmax with repeatability levels of agreement varying between +43% and −46% for all lesions taken together. For local recurrences, values vary between +23% and −33%; for the smaller lymph node and bone metastases, values are +40% to −49% and +56% to −47%, respectively. In addition to this, we show a significant correlation between lesion size and SUVmax repeatability levels of agreement.

With respect to our second aim, no significant differences in repeatability between standard OSEM reconstruction and BPL reconstruction could be shown in this pilot study. There was a small but not significant difference in favor of the standard OSEM reconstruction. However, BPL reconstructions resulted in significantly higher SUVmax of tumor lesions as compared to standard OSEM reconstructions, with significantly higher relative increases in smaller lesions.

Pollard et al. reported on the repeatability of Ga-68 PSMA-HBED-CC (PSMA-11) SUVmax in relapsing prostate cancer [20]. Repeatability levels of agreement were given for lymph node and bone metastases only and were lower than in our study: 30–40% versus 40–60%. Moreover, for other radiotracers like F-18 DCFPyL, F-18 FDG and F-18 FLT, values in the 30–40% range were reported for SUVmax [21,22]. A possible explanation for the higher day-to-day variability in our study is the relative high number of small lesions. Our study hints at a negative correlation between lesion SUVmax variability and lesion size (Figure 2A). We believe the patient cohort of this pilot study to be representative for patients with relapsing prostate cancer showing relatively small tumor lesions in lymph nodes and bones. Pollard et al. did not find any relationship between lesion size and SUVmax variability [20]. A possible explanation for this is that they only reported on relationships within classes, i.e., within typically smaller lymph node and bone metastases, where we report on all lesions including the larger relapses of the primary tumor in the prostate bed. Olde Heuvel et al. also reported on larger tumors in the setting of primary staging of prostate cancer and found smaller day-to-day variability [23].

A larger day-to-day variability may have implications for lesion-specific response in treatment monitoring. With respect to treatment response monitoring using F-18 FDG, Wahl et al. proposed a minimum of 30% SUVmax decrease for a true response in their PERCIST response criteria [24]. Although the PERCIST criteria have not been validated for Ga-68 PSMA PET yet, we think that a minimum of 30% response is probably not appropriate for the majority of the relatively small lesions that are typically found in (early) relapsing prostate cancer. Our study shows that a minimum response of 50% might be more appropriate in these cases, when using Ga-68 PSMA. Being deferred from a single center single vendor study, an even more conservative approach might be warranted when multiple centers and/or vendors are involved.

When confronted with the relatively small tumor lesions in patients with (early) relapsing prostate cancer, partial volume effects have to be anticipated. This will be particularly the case when quantifying tracer expression in small lesions using tracers with high tumor-to-background ratios like Ga-68 PSMA but is not always appreciated [25,26]. For example, in a report on the effects of androgen deprivation therapy on Ga-68 PSMA SUVmax in primary prostate cancer, lymph node and bone metastases as recent as April 2020, possible impact of partial volume effects was not discussed at all [27].

The possible importance of partial volume effects can be illustrated by the BPL versus standard reconstructions, as applied in our study. Two factors that contribute to the partial volume effect (spill-in and spill-out) are the finite spatial resolution of the imaging system (involving scanner hardware, acquisition parameters and reconstruction method) and image sampling on a discrete voxel grid imperfectly matching the actual contours of tracer distribution [12]. Impact of partial volume effects is strongly dependent on lesion size and is coming into play when lesion size falls below 2 to 3 times the resolution of the system, i.e., below 10 to 15 mm for an average PET/CT scanner system [10]. With respect to tracer SUVmax measurements, partial volume effect will generally result in lower SUVmax values. As SUVmean is usually defined as the average SUV in a 3D isocontour at a certain threshold set as a percentage of the SUVmax [28]. In standard OSEM reconstruction, the number of iterations is very limited, to prevent the emergence of excessive noise [13]. The Bayesian penalized likelihood (BPL) reconstruction algorithm called Q.Clear allows for improved spatial resolution because of its convergence, applying a noise penalty at the voxel level during image reconstruction [14]. The resulting higher uptake/expression values in small lesions (compared to standard reconstruction) have been reported for several radiotracers, including F-18 PSMA-1007 in prostate cancer patients [29]. In the latter study, SUVmax with BPL reconstruction has been compared to standard OSEM reconstruction, stratified for lesion size. A significant reported increase in SUVmax with BPL reconstruction for lesions smaller than 10 mm diameter only was reported. In our study, we confirm these findings for the ^68^Ga-PSMA tracer, including the relationship with lesion size (Table 2), thus highlighting the importance of partial volume effects and its correction.

We hypothesized that the BPL reconstruction algorithm Q.Clear has the ability to lower signal variability in the small lesions typically encountered in early relapse or early metastatic disease of prostate cancer. Our study did not confirm our hypothesis. On the contrary, signal variability tended to be higher with BPL reconstruction compared to standard reconstruction (although not significant). A possible explanation for this could be the ‘overshoot’ reported with BPL reconstruction for small spheres at high sphere-to-background ratios in a phantom study using F-18 FDG [30]. The high sphere-to-background ratios in this study may correspond to the high tumor-to-background ratios typically encountered in Ga-68 PSMA avid prostate cancer lesions.

Limitations. This single site study has several limitations. For a formal test–retest study, there is a relatively wide range in administered activity (0.9–1.7 MBq/kg) and time from injection to acquisition (55–69 min). However, in most patients, the activity was administered within reasonable limits with respect to dose and equilibration time, thus reflecting daily practice. Especially, with regard to relatively short-lived radiopharmaceuticals like Ga-68, exact dosing can be cumbersome. For example, timely administration of the radiopharmaceutical was hampered by the pending results of the quality control just prior to injection in some patients. The assignment of tumor lesions was only done by two experienced image readers, and inter-observer variability was not assessed. This was considered acceptable because the exact nature of the lesions was less important in light of the general aim of the study, i.e., assessment of signal variability. The same holds for the fact that for the majority of the lesions there has been no confirmation by histopathology. Moreover, potential detrimental effects of patient motion cannot be ruled out. This may be especially relevant because most lesions were small.

With respect to the comparison of the BPL and standard OSEM reconstructions, the image reading was not blinded. This was deemed acceptable because the visual appearance of both reconstructions is different, precluding true blinded image reading.

No significant difference was found between BPL and standard OSEM reconstruction signal variability, probably because the study was underpowered. Being a pilot study, the results can be used for power calculations for a possible future study.

## 5. Conclusions

The main finding of this study is the relatively high day-to-day variability of tumor SUVmax with repeatability levels of agreement varying between +43% and −46% for all lesions taken together. Small lesions tend to have larger day-to-day variability of tumor SUVmax when compared to larger lesions. With respect to response monitoring, minimum response of 50% for Ga-68 PSMA PET might be more appropriate.

With respect to our second aim no significant differences in repeatability between standard OSEM reconstruction and BPL reconstruction could be shown in this pilot study. There was a small but not significant difference in favor of the standard reconstruction, however. These results do not support the use of BPL with respect to our hypothesis that it might lower signal variability.

## Figures and Tables

**Figure 1 diagnostics-11-01100-f001:**
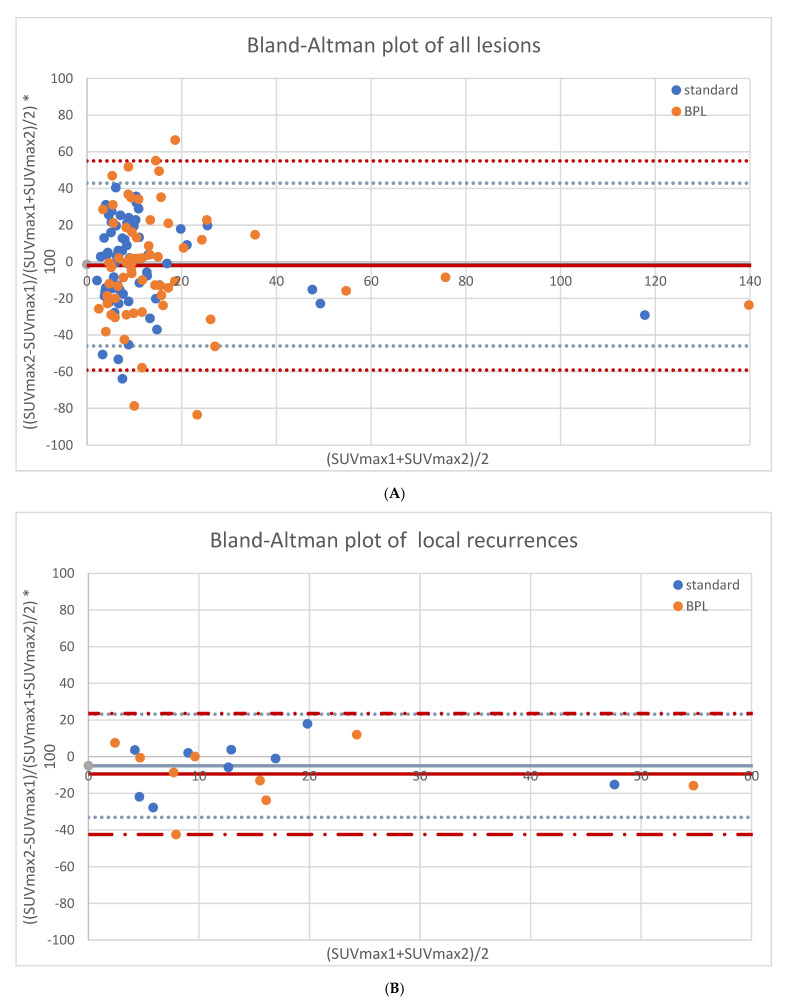
Repeatability results of SUVmax for both reconstructions in all lesions (65 lesions, (**A**)), local recurrences (9 lesions, (**B**)), lymph node metastases (36 lesions, (**C**)) and bone metastases (20 lesions, (**D**)).

**Figure 2 diagnostics-11-01100-f002:**
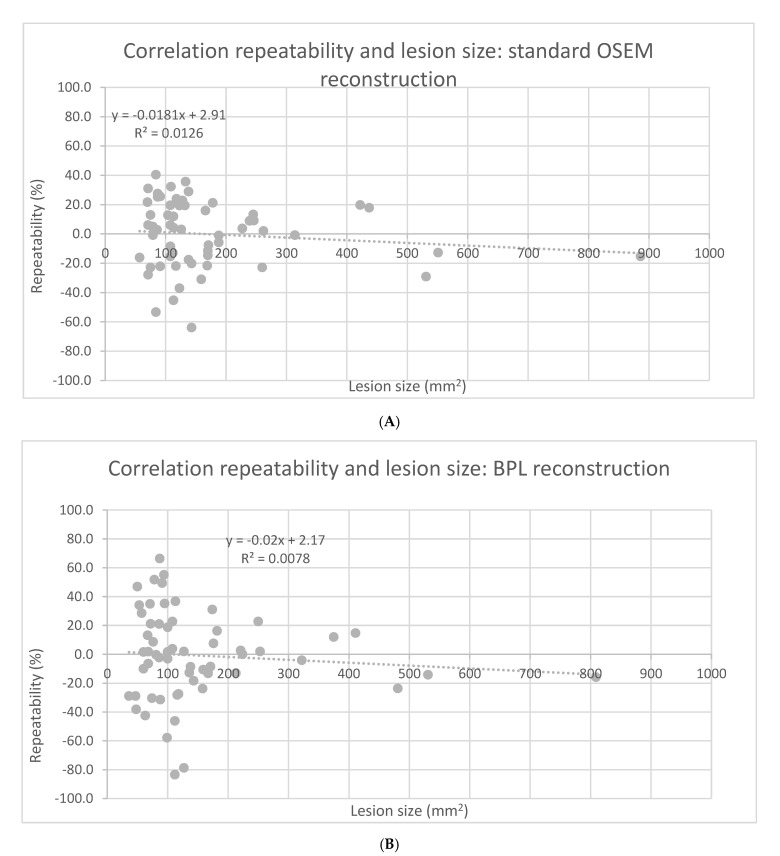
Correlation between SUVmax repeatability and lesion size for standard OSEM (**A**) and BPL reconstruction (**B**).

**Table 1 diagnostics-11-01100-t001:** Patient characteristics.

	Age	PSA	Gleason Score	Activity Test Scan (MBq/kg)	Activity Retest Scan (MBq/kg)	Time Test Scan (min)	Time Re-Test Scan (min)	Total Number of Lesions	Prostate Bed	Lymph Node Metastases	Bone	Initial Therapy	Year of Therapy
Pat no.													
1	83	2.4	7	1.2	1.4	60	65	3	-	3	-	LND +EBRT	2009
2	71	4.1	8	1.2	1.5	58	58	1	-	-	1	RALP +LND	2017
3	75	4.5	7	1.4	1.5	57	60	1	-	-	1	RALP	2009
4	73	8.4	6	1.3	1.6	57	57	6	-	3	3	AS+BT +LND	2012
5	69	0.7	7	1.3	0.9	55	56	2	-	2	-	RALP +ELND	2015
6	78	16.0	6	1.5	1.4	55	56	3	1	2	-	BT	2011
7	84	9.0	6	1.5	1.5	58	58	4	1	3	-	BT	2012
8	80	0.7	8	1.3	1.5	57	57	2	1	-	1	RALP +LND	2008
9	62	3.9	6	1.5	1.6	60	55	1	1	-	-	BT	2013
10	77	3.0	7	0.9	0.9	69	72	1	-	1	-	RALP +ELND	2010
11	71	3.5	7	1.3	1.5	55	56	1	1	-	-	BT	2014
12	67	5.7	-	1.3	1.1	55	55	3	1	1	1	BT	2007
13	78	1.7	8	1.4	1.4	61	56	1	-	1	-	BT	2016
14	75	2.8	6	1.4	1.4	55	55	3	-	3	-	BT	2009
15	74	2.0	7	1.5	1.5	59	62	1	-	1	-	RP+LND +EBRT	2009
16	77	1.2	7	1.0	1.5	55	55	3	-	3	-	RALP	2009
17	72	2.5	7	1.7	1.6	55	55	12	-	-	12	RALP +ELND	2018
18	73	2.8	7	1.7	1.4	58	58	1	-	-	1	BT	2007
19	78	5.4	6	1.5	1.4	57	55	3	-	3	-	RALP +LND	2008
20	77	3.7	8	1.5	1.3	55	55	4	2	2	-	EBRT +HT	2009
21	69	5.2	6	1.4	1.3	59	59	1	1	-	-	EBRT	2012
22	68	0.6	7	1.3	1.5	60	60	3	-	3	-	RALP +LND	2015
23	78	7.0	7	0.9	1.5	55	58	5	-	5	-	EBRT	2017

AS = active surveillance, BT = brachytherapy, EBRT = external beam radiotherapy, ELND = extended lymph node dissection, HT = hormonal therapy, LND = lymph node dissection, RP = radical prostatectomy (open procedure), RALP = robot assisted radical prostatectomy.

**Table 2 diagnostics-11-01100-t002:** BPL SUVmax increase (relative to standard reconstruction) for smaller and larger lesions.

	Test BPL Relative Increase of Suvmax (%)	SEM	Retest BPL Relative Increase of SUVmax (%)	SEM
lesions < 200 mm^2^	44.3	4.6	43.5	3.9
lesions ≥ 200 mm^2^	25.5	4.2	18.6	3.1
2-sided t-test	*p* = 0.004		*p* < 0.001	

## Data Availability

The data presented in this study are available on request from the corresponding author.

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
