# Peer review of "Evaluation of Quantitative Ga-68 PSMA PET/CT Repeatability of Recurrent Prostate Cancer Lesions Using Both OSEM and Bayesian Penalized Likelihood Reconstruction Algorithms"

_diagnostics, 2021, doi:10.3390/diagnostics11061100_

Round 1

Reviewer 1 Report

The authors present an interesting study evaluating the repeatability of Ga-68 PSMA for prostate cancer lesions using OSEM and BPL reconstruction algorithms in a single centre from a GE PET/CT scanner.  This is useful work which will be of help to the field.  They correctly point out the importance of imaging in following up specific lesions as opposed to the PSA measurements often done.

Major comments

Part of the paper is to consider the impact of BPL but there is no detail in the introduction about the BPL algorithm and previous work so this should be added.  They state that BPL corrects for partial volume effects (PVE) as part of BPL, whilst this is true OSEM+PSF also corrects for the PVE.  The difference is BPL runs to effective convergence whereas OSEM is not due to excess noise  (this causes increases in SUVmax for BPL).  Similarly they state in the discussion that the higher spatial variability could be due to partial volume correction in BPL, as I understand it the same PSF recon is used in BPL as for the GE OSEM PSF reconstruction (SharpIR). 

In addition detail should be added as to why the authors hypothesised that BPL would specifically lower the variation on SUVmax.  A regularised reconstruction such as BPL may have a smoothing impact on the SUVmean more than the SUVmax.

The details of the “standard reconstruction” need to be added to the methods eg iterations/subsets/filter.  This is required in order to compare the results to BPL where the beta value has been given (it is stated that another paper in this special issue evaluates the optimum beta).  The “standard reconstruction” includes PSF modelling which is probably not used as standard at all hospitals so it should be clear PSF  modelling is included.

Why have the x-values on the Bland-Altman plots been multiplied by 100?  Figure 1A has scale to 14,000 whereas 1B-D which are separated versions of 1A have scales to 6000

Potential patient movement should be added to the discussion

It is stated that this study “hints at a negative correlation” please could the p value be added to assess the significance of this correlation?

In conclusion it is stated that there was a small difference in favour of the standard reconstruction – was this significant?  If yes this should be stated, if not this should also be stated.

Minor comments

Abstract, add that this is a single centre study

Put “OSEM+PSF” rather than just OSEM.

Methods, include range for number of days test-retest scan

Make clear that the OSEM+TOF+PSF is subsequently referred to as “standard reconstruction” in the paper. 

What matrix size was used?

Were the OSEM and BPL images opened together when measuring the SUVmax to ensure the same lesions were measured each time?

Please include the data (perhaps as a supplementary table) when you measured the low dose CT to check the appropriateness of the lesion area in PET.  How many lesions was this done for?

Table 1 should have “activity” not “dose” 

Consider adding an example image to the paper

Use USA or UK English throughout

Use consistent font size

Reviewer 2 Report

The authors present data on the value of PSMA-PET test and retest scans in 23 patients with BCR before therapy initiation and report and the repeatability of SUV measurements with respect to standard OSEM and BPL algorithms. They found no statistical difference with regard to test-retest repeatability with both reconstruction methods. Also, repeatability was poorer in smaller lesions on both reconstruction methods.

Introduction:

Considering the inclusion of patients with BCR, the first half of the introduction is not useful in this context. The authors focus on the value of PSMA-PET for response assessment, however, this is not important for the BCR setting. This has to be changed and shortened. This is particularly interesting when considering that all patients in this cohort were therapy naïve. If the authors want to highlight the impact of repeatability of SUV measurements with respect to comparability, this has to made clearer. Currently this is not clear immediately. Also, please refrain from bold letters in the description of the study aims.

Methods:

How do the authors justify the rather low sample size (for both patients and lesions). I am aware that test-retest is a special situation, nonetheless 23 patients seems relatively low. Which sample size calculation method has been applied? Which distribution and differences were estimated?

Why did the authors not include patients in various clinical scenarios (e.g. primary staging, mCRPC, h/o hormone therapy and/or chemotherapy)?

Results:

The authors describe higher SUVmax values in BPL vs. OSEM. Was this also the case for biodistribution? Was there any difference in Tumor-to-Background ratios?

Discussion:

The authors discuss the impact of their data on response assessment using PERCIST criteria. Firstly, PERCIST has not been validated in PSMA-PET, yet. Thus such comparisons have to be interpreted with caution. Secondly, PET-based response assessment always considers PSMA-Uptake in mean tumor volume rather than single lesions (Recommendations by Fanti et al. EJNMMI 2021). This is not discussed at al. The effects of “size” will be lower considering whole tumor volume. The authors should add this analysis to their data. However, there will be the limitation of a highly preselected cohort (e.g. BCR with only few metastases or single lesions). This furthermore limits the impact on response assessment in general as the already available guidelines rather include patients with medium to high tumor load.

Furthermore, what is the scientific merit of this study? What do the authors imply with their data. If I interpret their data, I would think that BPL is useless. Something, which most sites do not apply currently anyway! The last chapter of the discussion should be adjusted to the question whether the authors data will improve PSMA imaging or not. Even in case of negative hypotheses (which is currently the case) the authors should still make a clear statement whether BPL is forth pursuing or not.

Round 2

Reviewer 1 Report

The authors have addressed the majority of my comments but two remain:

1 - I’d suggested the details (iterations/subsets/filter) of the OSEM reconstruction get added to the methods but only iterations/subsets have been.  Please could you add the filter choice too?  This is useful to know when comparing to other reconstruction algorithms.  Also there seems to be a typo in the text as it says 24 iterations and 2 subsets, should this be 2 iterations 24 subsets which is the GE default reconstruction?  Please could the authors add a comment for why they used this reconstruction with PSF.  Generally when PSF modelling is included an extra iteration is used compared to OSEM without PSF if they did use 2 iterations. 

2 - Please include the range for number of days for the test-retest as previously suggested.  You could add the fact that only one patient had 23 days between scans and the next highest was xx days as you are concerned about including the fact one patient had 23 days between scans.

Also, the authors have added in this revision “these results do not support the use of BPL in this clinical situation”.  This should be clarified to just refer to their second hypothesis that BPL would reduce signal variability.  The authors haven’t examined the use of BPL for imaging prostate cancer itself just the repeatability of the measurements.

Author Response

The authors have addressed the majority of my comments but two remain:

1 - I’d suggested the details (iterations/subsets/filter) of the OSEM reconstruction get added to the methods but only iterations/subsets have been.  Please could you add the filter choice too?  This is useful to know when comparing to other reconstruction algorithms.  Also there seems to be a typo in the text as it says 24 iterations and 2 subsets, should this be 2 iterations 24 subsets which is the GE default reconstruction?  Please could the authors add a comment for why they used this reconstruction with PSF.  Generally when PSF modelling is included an extra iteration is used compared to OSEM without PSF if they did use 2 iterations. 

We thank this reviewer for the further comments. We have added the filter choice and corrected for the typo. As far as we know only 2 iterations have been used in OSEM+PSF, so no extra iteration there.

2 - Please include the range for number of days for the test-retest as previously suggested.  You could add the fact that only one patient had 23 days between scans and the next highest was xx days as you are concerned about including the fact one patient had 23 days between scans.

This has been changed according to reviewer’s suggestion.

Also, the authors have added in this revision “these results do not support the use of BPL in this clinical situation”.  This should be clarified to just refer to their second hypothesis that BPL would reduce signal variability.  The authors haven’t examined the use of BPL for imaging prostate cancer itself just the repeatability of the measurements.

This has been changed according to reviewer’s suggestion, too.